# Nuclear Factor of Activated T Cells-5 Regulates Notochord Lumenogenesis in Chordate Larval Development

**DOI:** 10.3390/ijms232214407

**Published:** 2022-11-19

**Authors:** Muchun He, Jiankai Wei, Yuting Li, Bo Dong

**Affiliations:** 1Sars-Fang Centre, MoE Key Laboratory of Marine Genetics and Breeding, College of Marine Life Sciences, Ocean University of China, Qingdao 266003, China; 2Laoshan Laboratory, Qingdao 266237, China; 3Institute of Evolution and Marine Biodiversity, Ocean University of China, Qingdao 266003, China

**Keywords:** *NFAT5*, notochord, lumen formation, osmotic stress, ascidian

## Abstract

Osmoregulation is essential for organisms to adapt to the exterior environment and plays an important role in embryonic organogenesis. Tubular organ formation usually involves a hyperosmotic lumen environment. The mechanisms of how the cells respond and regulate lumen formation remain largely unknown. Here, we reported that the nuclear factor of activated T cells-5 (*NFAT5*), the only transcription factor in the *NFAT* family involved in the cellular responses to hypertonic stress, regulated notochord lumen formation in chordate *Ciona*. *Ciona NFAT5* (*Ci*-*NFAT5*) was expressed in notochord, and its expression level increased during notochord lumen formation and expansion. Knockout and expression of the dominant negative of *NFAT5* in *Ciona* embryos resulted in the failure of notochord lumen expansion. We further demonstrated that the *Ci*-NFAT5 transferred from the cytoplasm into nuclei in HeLa cells under the hyperosmotic medium, indicating *Ci-NFAT5* can respond the hypertonicity. To reveal the underly mechanisms, we predicted potential downstream genes of *Ci-NFAT5* and further validated *Ci-NFAT5-*interacted genes by the luciferase assay. The results showed that *Ci-*NFAT5 promoted *SLC26A6* expression. Furthermore, expression of a transport inactivity mutant of *SLC26A6* (L421P) in notochord led to the failure of lumen expansion, phenocopying that of *Ci-NFAT5* knockout. These results suggest that *Ci*-*NFAT5* regulates notochord lumen expansion via the SLC26A6 axis. Taken together, our results reveal that the chordate *NFAT5* responds to hypertonic stress and regulates lumen osmotic pressure via an ion channel pathway on luminal organ formation.

## 1. Introduction

Biological tubes are present in all multicellular animals, serving diverse physiological functions, including transportation of nutrients, waste and gases, and structural support. One of the basic design principles of biological tubes is the formation of a de novo lumen, which includes three steps: cell-matrix and cell-cell recognition, apical-basal polarization, and expansion of the luminal space by fluid or ion efflux. Once lumens are formed, the tubes expand to a mature functional size [1]. In luminal organ expansion, hydrostatic pressure is thought to be the dominating driving force. In the cardiovascular system, hypertension is the most common preventable risk factor for cardiovascular disease and death and is a growing health burden [2]. Hypertension leads to vascular phenotypes characterized by arterial remodeling, endothelial dysfunction, reduction of elasticity, and an increase in vascular tone [3]. In computational models of lumenogenesis, hydrostatic pressure formed by the equilibrium of osmotic pressure, fluid influx, and paracellular leak contributes to lumen growth and homeostasis [4,5]. Changes in the apical delivery of ions by pumps and channels would create an osmotic pressure within the lumen of acini [1]. Ion channels, such as the cystic fibrosis transmembrane conductance regulator (CFTR) and Na-K-ATPase, are used for lumen expansion. In the absence of these channels, the lumen is greatly reduced in diameter. Pharmacological hyper-activation of the CFTR-dependent fluid transport results in overexpansion of the gut lumen in developing zebrafish [6] and MDCK cysts [7]. In the gut, loss of the *Tcf2* transcription factor strongly inhibits the expression of Na-K-ATPase and Claudin-15, leading to multi-lumen [8]. In *Caenorhabditis elegans* excretory cells, the water channel AQP-8 mediates transluminal flux, thereby influencing tube size [9]. The secretory machinery also has a role in lumen expansion. Bidirectional ER-Golgi transport is a key regulator of apical transport and lumen morphogenesis. The rapid bursts of COPI can secrete various molecules and fluids into the lumen [10,11].

Among the different forms of the *NFAT* family, *NFAT5/TonEBP* is the only member that regulates osmotic pressure-induced gene expression in invertebrates and mammalian animals [12]. Under high osmotic pressure in mammalian cells, *NFAT5* is synthesized and accumulated in the nucleus [13]. Its activity and localization are independent of calcineurin-mediated dephosphorylation [14,15], and increased *NFAT5* transcription is associated with p38 MAPK-mediated phosphorylation [16]. To restore biochemical homeostasis under hypertonic stress, cells induce a genetic program of osmotic adaptation responses, in which the intracellular electrolyte is gradually replaced by small non-charged organic permeates [13]. The production and uptake of organic by the sodium/myo-inositol cotransporter (*SMIT*) [17], sodium chloride/betaine cotransporter (*BGT1*) [18,19], sodium chloride/taurine cotransporter (*TauT*) [20], aldose reductase (*AR*, for the biosynthesis of sorbitol) [21], and aquaporin channels (*AQP1* and *AQP2*) [22]. The expression of these genes has been reported to be regulated by NFAT5.

*Ciona* belongs to the urochordate, which is the closest relative of vertebrates [23]. The notochord tubulogenesis of *Ciona* is a cord hollowing process [24] that includes complex morphological events: cell elongation, lumen formation, cell migration, and lumen fusion [25,26]. An extracellular pocket lumen is formed between two adjacent cells, and eventually, the pocket lumens coalesce into a single lumen in notochord [27]. Ion channels such as *SLC26-2* and apical protein secretion were reported to participate in lumen formation, suggesting that the osmotic pressure plays a role during notochord lumen expansion. However, the mechanism of how notochord senses osmotic pressure and thus regulates lumen expansion is unclear.

In *Ciona*, *NFAT5* has been reported as the downstream gene of *brachyury* and expressed in notochord cells [28]. Here, we reported that *Ciona NFAT5* (*Ci*-*NFAT5*), a transcription factor response to hypertonicity, was required for *Ciona* lumen expansion. We further demonstrated that *SLC26A6*, an Slc26 family anion transporter, was regulated by *Ci-NFAT5*. Overexpressing *SLC26A6* (L421P) arrested lumen expansion, phenocopying that of *Ci-NFAT5* knockout. Taken together, our results reveal that *Ciona NFAT5* (*Ci-NFAT5*) can respond to hypertonicity and regulates notochord lumen formation and expansion via ion channel *SLC26A6*.

## 2. Results

### 2.1. Phylogenetic and Expression Pattern Analysis of NFAT5

*Ciona NFAT* family genes were identified using blast against vertebrate *NFAT* sequences. Only one *NFAT5* gene was acquired, and no significant hit results for *NFAT1-4* were obtained in the *Ciona* genome. To understand the relationship and origin of *Ciona*-NFAT5, we downloaded NFAT protein sequences of 12 species from the public database (Appendix A) and constructed a phylogenetic tree (Figure 1A). The results showed that there are two clades: NFAT1-4 were clustered together, while NFAT5 proteins were clustered in another clade. NFAT1-4 proteins have calcineurin binding domain and were identified only in vertebrate animals, while NFAT5 proteins have no calcineurin binding domain (Appendix A) and were identified in both invertebrate and vertebrate [29]. Searching conserved domains within NFAT5 proteins identified the conserved RHD (DNA binding domain) and IPT (Dimerization domain) among all NFAT5 sequences (Figure 1B). The multiple DNA binding domain (DBD) sequence alignments showed that the DBD domains were highly conservative (Appendix A).

We explored the expression profiles of *Ci-NFAT5* at different embryonic and larval stages (0, 5, 10, 14, 18, 21, 31, 42 h postfertilization, hpf) by qRT-PCR. The results showed that the expression level of *Ci-NFAT5* increased from 14 to 21 hpf when the notochord lumen started to format and expand, then decreased from 31 to 42 hpf (Figure 1C), indicating that *NFAT5* plays roles during lumen formation. Then, to determine the tissue expression pattern of *Ci-NFAT5*, we made an *NFAT5* promoter construct, which included 3054 bp upstream of *NFAT5* and expressed it in *Ciona* embryos. The fluorescence was observed to be expressed in the notochord at the lumen formation stage (Figure 1D).

### 2.2. Loss of NFAT5 Resulted in the Failure of Lumen Formation and Expansion

To determine the roles of *Ci-NFAT5* in notochord lumen formation, we generated a C-terminal deletion construct of *Ciona* NFAT5 fusion proteins (*Ciona*-NFAT5 DN), which retains DNA binding and dimerization domains, and lack the transactivation domain, serving as a dominant negative (DN) mutant (Figure 2A). This mutant construct has been reported to diminish hyperosmotic induction of the downstream gene in a dominant negative manner [30,31]. *Ciona*−NFAT5 DN construct was electroporated into the fertilized eggs with a notochord−specific *brachyury* promoter to force its expression in the notochord cells. The results showed that overexpression of *Ciona*-NFAT5 DN prevented notochord lumen formation (Figure 2B).

To confirm the phenotypes, the sgRNA was designed for the *Ci*-*NFAT5* knockout by CRISPR/Cas9 (Figure 2C). We extracted the DNA from the gene-edited embryos and cloned the 386 bp segment around the editing site. Then, T7 Endonuclease was used to examine the editing efficiency. The results showed that the mutation efficiency was 20.69% of sgRNA by sequencing (Appendix A). The mutant notochord showed defects in lumen formation compared to the control embryos. The apical membrane of the notochord bulged into the lumen space in knockout animals (Figure 2D). Both dominant negative overexpression and knockout results indicated that *Ci*-*NFAT5* is required for notochord lumen formation and expansion.

### 2.3. Ciona-NFAT5 Responds to Hypertonicity

In order to reveal the mechanisms of *Ci*-NFAT5 on notochord lumen formation and expansion in *Ciona* embryos, we examined its response capability to the osmotic pressure using culture cell lines. The medium with different osmotic pressure was prepared by adding D-mannitol to the basic medium (Appendix A). The results showed that the nucleus/cytoplasm ratio of *Ci*-NFAT5 fluorescence increased significantly after hypertonic treatment 11 h (653 mOsm/kg), indicating that *Ci*-NFAT5 can respond to osmotic pressure variation (Figure 3A,B). Furthermore, we observed the distribution of *Ci*-NFAT5 in notochord cells. The results showed that with the lumen expansion, signaling intensity in the nucleus became stronger gradually (Figure 3E). These results indicated that *Ciona*-NFAT5 could respond to hypertonicity, similar to *Homo*-NFAT5 (Appendix A).

### 2.4. Ci-NFAT5 Regulates Lumen Expansion via SLC26A6 Pathway

To identify the target genes of *Ci-NFAT5*, we performed ATAC-seq (transposase-accessible chromatin with high-throughput sequencing) and identified 23,037 open sites at the lumen expansion stage (22 hpf). In the biological process category of GO enrichment, the top 20 pathways were mainly related to transmembrane transport, transmembrane transport activity, and copper ion binding (Appendix A). These genes were crossed with previously reported notochord-specifically expressed genes [32], and 30 remained. We then screened the downstream genes using motif binding prediction (MEME software) (Appendix A). We further screened these genes through the promoter assay and found that *SLC26A6* expressed in the notochord tissue during lumen formation and expansion stages (Figure 4A). Moreover, we experimentally confirmed that *SLC26A6* was the NFAT5 downstream gene through the luciferase report gene assay. The expression levels of *SLC26A6* increased significantly compared to the control (Figure 4B,C). In addition, we noticed that the expression level change of *SLC26A6* was similar to *NFAT5* at 10, 14, and 21 hpf (Appendix A), indicating the regulatory relationship between them. To explore whether *Ci-NFAT5* affects luminal distension through *SLC26A6*, we generated a loss of transport activity mutant of SLC26A6 by substituting leucine to proline at position 421 (L421P) [33,34] (Appendix A) and forced it to be expressed in notochord by notochord-specific brachyury. The mutant overexpressing notochord cells showed lumen expansion defects, phenocopying with the knockout of *NFAT5* (Figure 4D). These results suggest that *SLC26A6* is regulated by *Ciona*-*NFAT5* and required for notochord lumen expansion.

## 3. Discussion

The body’s osmotic pressure is one of the most tightly controlled physiological parameters, regulated by the balance of hydration and solute concentrations [35]. The development of many tissues is sensitive to osmotic fluctuations, such as the kidney [21], eye lens [36], and intervertebral discs [37]. In closed 3D cellular systems, ion gradients can generate osmotic pressure, which can result in increased cellular tension, and then drive morphogenesis and maintain homeostasis [38]. Changes in the osmotic pressure of the extracellular fluid (ECF) cause water to flow across the cell membrane to balance the osmolality of the cytoplasm with ECF [39]. The large changes in ECF osmotic pressure can affect the physical integrity of cells and tissues [40] and the biological activity of life-sustaining macromolecules [41]. The renal medulla is the only hypertonic tissue in mammals under physiological conditions due to the operation of the urine concentration mechanism [19].

In *Ciona*, as lumens between notochord cells began to form, *SLC26-2* was observed only in the membrane lining [27]. *SLC26-2* is an ion transporter with diverse substrate specificity [42]. *Ciona*-*Slc26aα* (*Ci*-*Slc26aα*) is also an Slc26 family anion transporter. Adjacent notochord cells in *Ci*-*Slc26aα* knockdown embryos developed only narrow pockets of lumens that slowly acquired a crescent shape [34]. Not only ion transport but also apical protein secretion was involved in lumen formation [43]. Knockdown of *Caveolin-a* in *Ciona* embryos causes the failure of notochord elongation and lumenogenesis [44]. *14-3-3ea* and *ERM* also play a key role in regulating the early steps of tubulogenesis in *Ciona* embryos [45,46]. These studies indicate that ion channels and vesicle transport play an important role in tubular organ formation. These pathways may create a hypertonic environment that favors the expansion of the lumen in *Ciona*.

Our results also showed that changes in osmotic pressure affect the localization of *the Ciona*-NFAT5 protein in HeLa. *Ciona*-NFAT5 has a similar function to *Homo*-NFAT5, which responds to hyperosmotic stress. In *Ciona* notochord with the lumen expansion, the signaling intensity of NFAT5 in the nucleus became stronger gradually, indicating the lumen between notochord cells was hydrostatic and sticky in *Ciona*. During the stage of *Ciona* lumen formation, the dominant negative and knockout of *NFAT5* resulted in abnormal lumen formation. The apical membrane of notochord cells, when knockdown or knockout of *NFAT5*, bulged into the lumen (Figure 5).

In this study, we predicted downstream genes of NFAT5; NFAT5 promotes the expression of *SLC26A6*. In many tubular structures, transepithelial ion transport is critical for the secretion and absorption of luminal fluid. SLC26A6 has a conserved domain as all SLC26 family members, with a sulfate transporter domain in the transmembrane region and a sulfate transporter and anti-sigma factor antagonist (STAS) domain in the C terminus (Appendix A). *SLC26A*6 has the similar function with *SLC26Aα*, which can mediate the transport of Cl^−^/HCO_3_^−^, Cl^−^/formate, Cl^−^/oxalate, Cl^−^/nitrate, SO_4_^2−^/oxalate and Cl^−^/OH^−^ [47]. *Ci*-*SLC26Aα* is initially involved in providing an osmotic drive for water influx to inflate the notochord lumen [34]. Secreted HCO_3_^−^ binds to luminal Ca2^+^ to form CaCO_3_ precipitates, resulting in a 70–100 mOsmol/kg H_2_O reduction in luminal osmolality [48,49]. In this study, we employed both in vitro and in vivo approaches to understand the function of *NFAT5* from the invertebrate chordate *Ciona* during embryonic lumen formation. Our results demonstrated that the invertebrate chordate *NFAT5* responds to hyperosmotic stress and regulates notochord lumen expansion via the SLC26A6 pathway (Figure 5).

## 4. Materials and Methods

### 4.1. Experimental Animals

Adults of *C. robusta* were collected from Rongcheng harbor bay, Weihai, China. The animals were maintained in the 32‰ salinity of seawater with continuous light in the laboratory. We dissected adult animals to obtain eggs and sperm. After fertilization and dechorionation, eggs were electroporated, and then the embryos were cultured at 16 °C, 18 °C, or 24 °C. Embryos at different stages were collected for RNA extraction and morphological observation.

### 4.2. Electroporation

Electroporation was followed according to the method [50] described previously with some modifications. 60–80 μg of plasmids in 80 μL final volume adjusted with ddH_2_O and was electroporated with 420 μL of 0.77 M D-Mannitol.

After fertilization, the dechorionated eggs (300 μL) were mixed using a Gene Pulser Xcell System (BIO-RAD) in 0.4 cm cuvettes. The exponential protocol was used with 50 V and 1500 μF as a parameter. Once electroporated, the fertilized eggs were washed and cultured at 16 °C or 18 °C.

### 4.3. Phylogenetic and Motif Analysis

NFAT5 protein sequences of these species were downloaded from the NCBI databases (Appendix A). The phylogenetic tree was constructed using the Maximum-Likelihood (ML) method by the Molecular Evolutionary Genetic Analysis of MEGA X. The multiple sequence alignment was performed using CLC Main Workbench 5.

### 4.4. Quantitative Real-Time PCR (qPCR)

Total RNA was extracted from different stages of frozen embryos using RNAiso plus (TAKARA, Japan). The integrity and quality of total RNA were determined by agarose gel electrophoresis and Nanodrop spectrophotometry (Thermo Fisher). The reverse transcription was carried out using 1 μg total RNA from different stages of embryos by HiScript II Q RT SuperMix for qPCR (+gDNA wiper) (Vazyme, Nanjing, China). The expression analysis of *NFAT5* and *SLC26A6* at different stages was analyzed by 2-step qPCR (Vazyme, Nanjing, China) using a Light Cycler 96 (Roche, Basel, Switzerland). The reaction condition was as follows: 95 °C for 30 s, 40 cycles at 95 °C for 10 s and 60 °C for 30 s, 95 °C for 15 s, 60 °C for 60 s, and 95 °C for 15 s. The expression level of *NFAT5* was normalized using Tubulin as a reference. Data were calculated using the 2^−ΔΔCt^ method, and statistical analyses were performed using paired Student’s t-tests. *p* < 0.05 was considered statistically significant. The graph of qPCR results was plotted using Prism 9 software.

### 4.5. Plasmid Constructions

Full length of cDNA encoding *Homo*-, *Ciona*-*NFAT5* were amplified via PCR reaction. These two PCR products and pEGFP-N1 vectors were digested with both Xho1 and Apa1, Xho1 and BamH1, respectively. The digested vectors and fragments were then ligated to acquire the construction: CMV>*Homo*-NFAT5::eGFP and CMV>*Ciona*-NFAT5::eGFP. The clones were then sequenced in GENEWIZ company, and the corrected clones were used for the sequent experiments.

To create a notochord specific expression vector, a 1 kb promoter from the upstream region of the *brachyury* gene was amplified by the PCR method and then subcloned into Kpn1 and BamH1 sites to construct an expression vector *Bra*>EGFP. The dominant negative sequences were amplified by the PCR method. *Bra*>EGFP vectors were digested with BamH1. The digested vectors and fragments were then ligated to acquire the construction: *Bra*1k>*Ciona*-NFAT5 DN::eGFP.

Two target sequences of CRISPR/Cas9 against *Ciona* NFAT5 were designed by CRISPR direct (http://crispr.dbcls.jp, accessed on 19 November 2018). The sequence of selected single guide RNAs (sgRNAs) is AACACACCAGCATTCCAAAA, and the sequence of control sgRNA was designed according to the previous study [51]. Based on the target sequences of CRISPR/Cas9, sgRNAs were synthesized and cloned into the vector of Cr-U6>sgRNA(F + E) (Addgene number: 59986), respectively.

The 3 kb DNA sequence upstream of *NFAT5* and the predicted downstream genes from *Ciona* was amplified by PCR and subcloned into Nhe1 and ECoR1 restriction sites of the pCRE-Luc vector for luciferase analysis.

The 3 kb DNA sequence upstream of *NFAT5* and the predicted downstream genes from *Ciona* was amplified by PCR and subcloned into Kpn1 and Xho1 restriction sites of the pEGFP-N1 vector for promoter analysis.

All constructs were generated using the one ClonExpress MultiS One Step Cloning Kit (Vazyme, Nanjing, China) and confirmed by sequencing.

The sequences of primers used for plasmid construction and qPCR are shown in Appendix A.

The validation of the efficiency of the gRNAs was performed as per the previously described protocol [52].

### 4.6. Cells Culture and Transfection

HeLa cells were provided by Professor Liu Chenguang’s laboratory (Ocean University of China, Qingdao, China). HEK293T cells were obtained from Professor Guo Huarong (Ocean University of China, Qingdao, China). HeLa and HEK293T cells were cultured with DMEM medium. HeLa cells were incubated in a medium supplemented with 10% FBS, HEK293T cells with 15% FBS, 100 U/mL penicillin, and 100 mg/mL streptomycin, and were maintained in 5% CO_2_ at 37 °C as described previously [53]. Cells were subcultured for further experiments after they became 80–90% confluent. Transfection in HLEA cells was performed using Lipofectamine 3000 Invitrogen (Thermo Fisher). Transfection in HEK293T cells was performed using Hieff Trans™ Liposomal Transfection Reagent (Yeasen).

### 4.7. Hypertonic Treatment

The medium with different osmotic pressure was prepared by adding D-mannitol to the basic medium. The osmotic pressure of basic medium and 0.3 M D-Mannitol are 339 and 653 mOsm/kg, measured by algeoscopy. Hypertonic treatment was performed 48 h after transfection. After hypertonic treatment, cell culture slides with cells were fixed with 4% paraformaldehyde in PBS for 30 min at room temperature. Incubation with 1/300 FITC Phalloidin (Yeasen) in PBS. After washing, the cell culture slides were mounted in VECTASHIELD with DAPI (Vector Laboratories) before microscopic observation.

### 4.8. Double Luciferase Reporting System Experiment

For the luciferase assay, HEK293T cells were passaged in 96-well plates with 3 × 10^4^ cells per well. Then, the cells were co-transfected with 100 ng of downstream genes luciferase reporter plasmid, 10 ng of PRH-TK, and 100 ng of *CMV*>*Ciona*-NFAT5::GFP. Twelve hours after transfection, a fresh medium was placed into each well. Forty-eight hours after transfection, firefly and Renilla luciferase activities were measured using the Dual-Luciferase reporter assay system (Beyotime, China), and relative reporter activity was normalized to Renilla luciferase activity.

### 4.9. ATAC-Seq Analysis and Downstream Gene Screening of Ciona-NFAT5

To obtain ascidian embryos (22hpf), incubate eggs with activated sperm for 8 min and keep developing embryos at 16 °C in ASWH in a dish coated with a thin layer of 1% agarose melted in ASWH. We recommend starting ATAC-seq experiments with 60,000–80,000 cells. The process of ATAC-seq library preparation includes cell lysis, tagmentation, and DNA library sequencing. Raw reads were stored in FASTQ file format. The sequencing adaptors and poor-quality reads were removed by the Fastp software (v0.22.0) to obtain high-quality clean reads. Filtered reads were mapped to the reference genome using Bowtie2 (v2.4.4). MACS2 (v2.2.7.1) was adopted for peak calling. Gene Ontology (GO) enrichment analyses were performed using the clusterProfiler packages (v4.4.4) with the parameters (pvalueCutoff = 1) to determine if the genes are enriched for specific terms. Thirty genes were identified after cross ATAC-Seq and notochord cell RNA-Seq data. The motif was examined using the Multiple EM for Motif Elicitation (MEME) suite, and transcription factors were predicted using the motif database scanning algorithm FIMO (*p*-value ≤ 0.001).

## Figures and Tables

**Figure 1 ijms-23-14407-f001:**
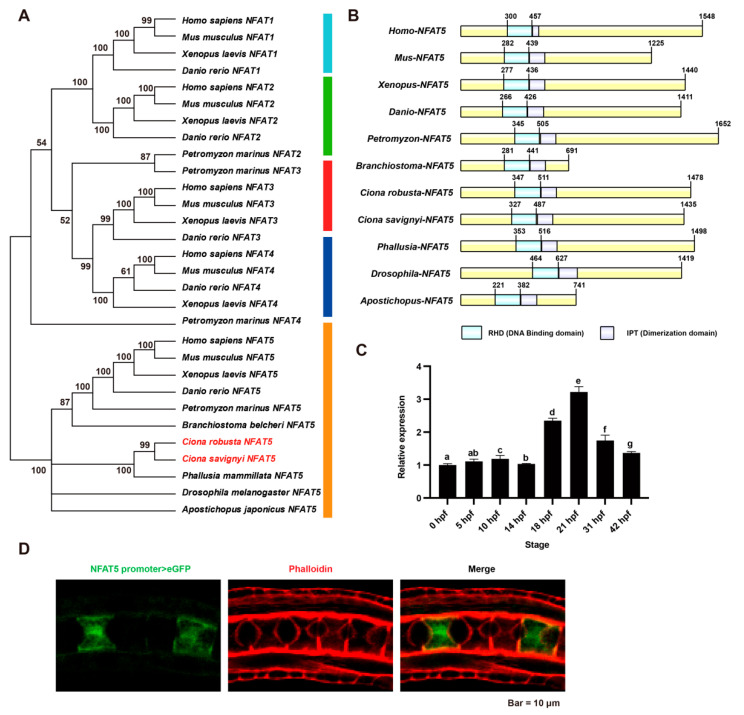
Evolutionarily analysis, the conserved domain alignment, mRNA expression profile analysis, and tissue expression pattern of *NFAT5*. (**A**) The phylogenetic tree of NFAT family genes was constructed by the maximum likelihood method. The LG model for amino acid substitution was used. The color strip denotes the cluster of NFAT1, NFAT2, NFAT3, NFAT4, and NFAT5. The red font highlighted *Ciona* NFAT5. (**B**) Schematic diagrams of the domain organization of all NFAT5 sequences. Light blue and light purple indicate conserved RHD (DNA binding domain) and IPT (Dimerization), respectively. (**C**) Expression profile of *Ci-NFAT5* from 0 to 42 hpf. There appears to be a high expression level at 18 and 21 hpf when the lumen is expanded. Significance difference is marked by letters (a–g). The identical letter indicates that the difference is not significant, whereas the different letter indicates the difference is significant. (**D**) The tissue expression pattern of NFAT5 by promoter assay. The fluorescence was observed in the notochord at the lumen expansion stage.

**Figure 2 ijms-23-14407-f002:**
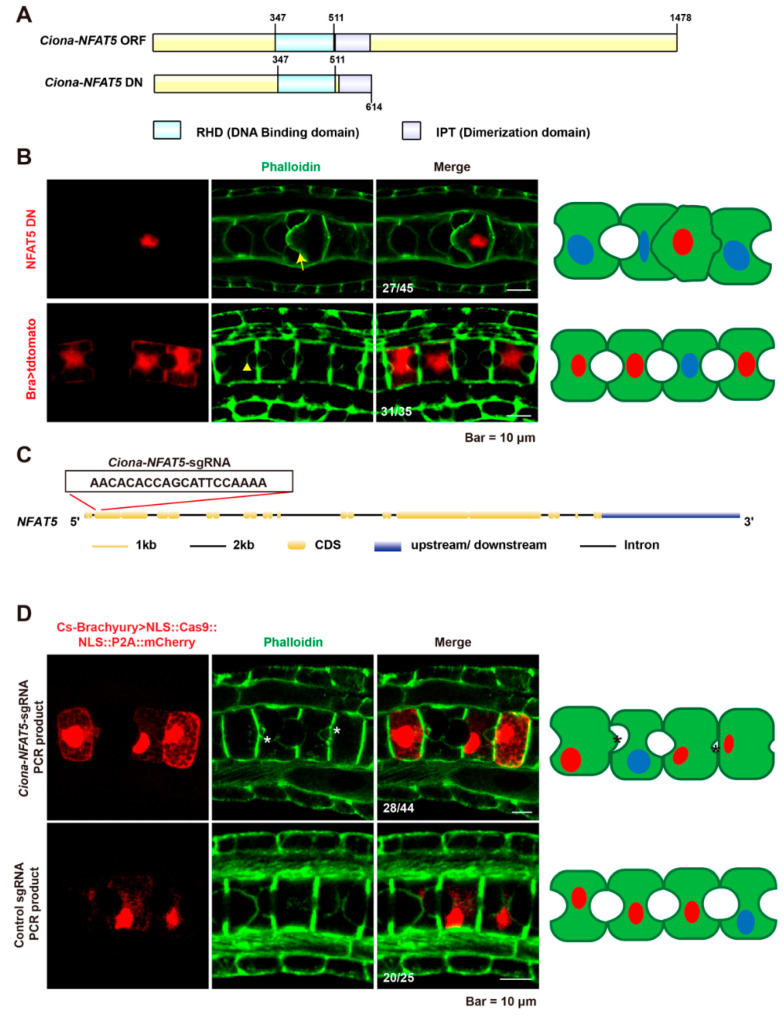
Loss of NFAT5 function resulted in disruption of lumen formation and expansion. (**A**) Schematic diagrams of the domains in wild type and dominant negative format of *Ciona*-NFAT5. (**B**) Overexpression of dominant negative *Ciona* NFAT5 led to the failure of lumen formation. (**C**) The sequences and the targeted site of single guide RNA (sgRNA) for *Ci-NFAT5* knockout by CRISPR-Cas9. (**D**) Confocal images of *Ciona*-*NFAT5* knockout embryos and control ones at the late tailbud stage. Asterisk represents defects on notochord lumen expansion. The ratio of embryos with phenotypes was labeled. The schematic images on the right side of the confocal images described the phenotypes of the *Ciona*-*NFAT5* knockdown/out and control. The green, blue, and red areas represented the notochord cells, the nuclei, and the nuclei in cells expressed in *Bra*>NFAT5-DN::tdtomato/*Bra*>NLS:Cas9:NLS:P2A:mCherry (red), respectively. The scale bar represents 10 μm.

**Figure 3 ijms-23-14407-f003:**
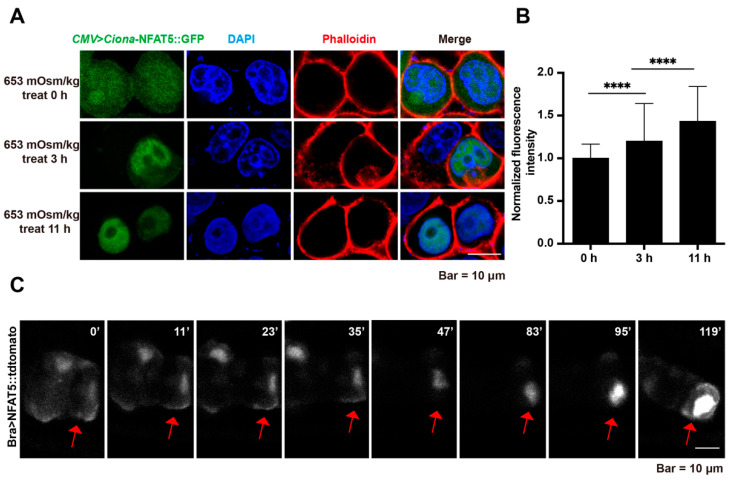
*Ciona*-NFAT5 responds to hypertonicity. (**A**) HeLa cells were transfected with *Ciona*-NFAT5. Representative cell images at 0, 3, and 11 h after hypertonic treatment (653 mOsm/kg). (**B**) Quantification of *Ciona*-NFAT5 translocation from the cytoplasm to the nucleus upon hypertonic treatment. The fluorescence intensity of the nucleus/cytoplasm was measured by image J. Asterisks (****) represented statistical significance (*p* < 0.0001). (**C**) Subcellular dynamics of *Ciona*-NFAT5 in *Ciona* notochord cells. The scale bar represents 10 μm.

**Figure 4 ijms-23-14407-f004:**
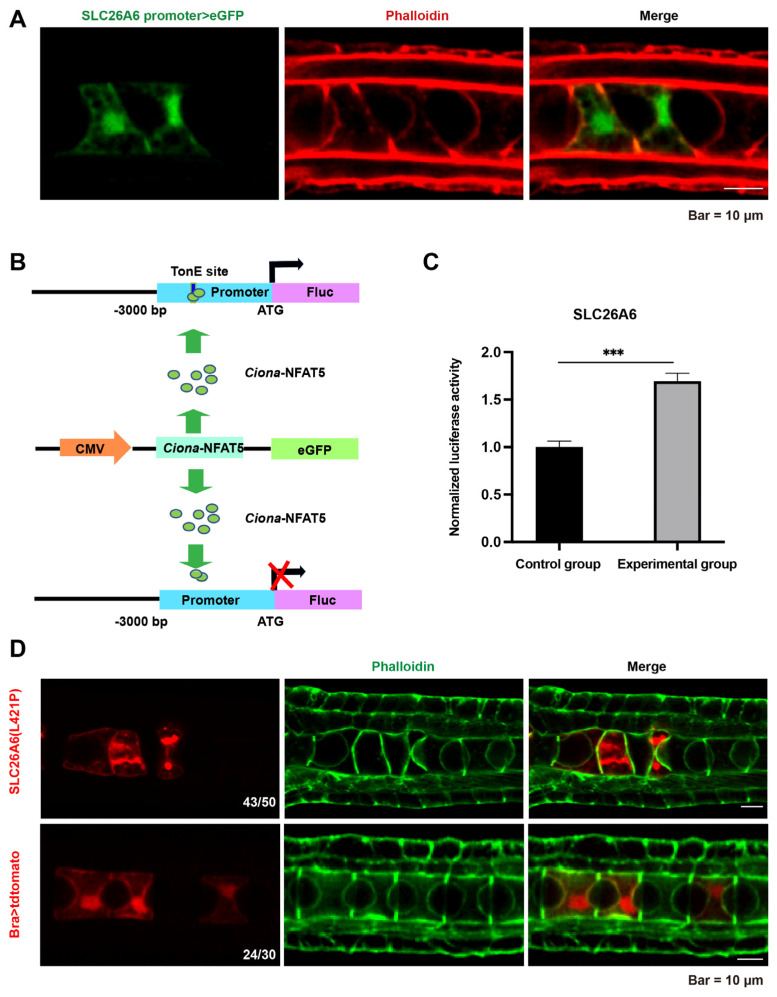
*Ciona*-*NFAT5* regulates notochord lumen expansion via *SLC26A6*. (**A**) The tissue expression pattern assay by *SLC26A6* promoter. The scale bar represents 25 μm. (**B**) The experimental diagram of the luciferase report gene assay. Schematic representation of luciferase reporter constructs carrying the 5′-UTR regions of downstream genes. The upstream 3 kb DNA sequence of the predicted downstream genes from *Ciona* was subcloned into the pCRE-Luc vector for luciferase analysis. (**C**) Relative luciferase activity in HEK-293T cells co-transfected with 5′-UTR *SLC26A6*, *NFAT5* ORF, and PRH-TK, respectively. Firefly luciferase values were normalized Renilla luciferase activity. Student’s *t*-test was used to evaluate the significance of luciferase data. Asterisks (***) represent statistical significance (*p* < 0.001). (**D**) Overexpression mutant of *SLC26A6* in *Ciona* notochord caused lumen expansion defects. The scale bar represents 10 μm.

**Figure 5 ijms-23-14407-f005:**
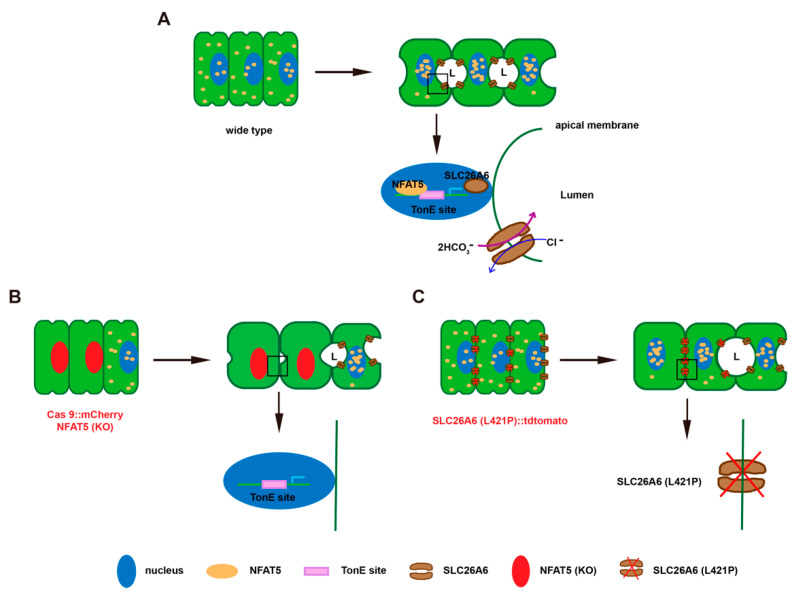
A working model of the regulatory roles of *Ciona*-*NFAT5* and *SLC26A6* on ascidian notochord lumen expansion. (**A**) The schematic diagram depicts that *NFAT5* responds to hyperosmotic stress and regulates notochord lumen expansion via the *SLC26A6* pathway. (**B**) Loss of NFAT5 function in notochord cells by CRISPR/Cas9 suppressed the expression of the downstream, genes including SLC26A6, and led to the defects of lumen expansion. (**C**) Loss of transport activity mutant of SLC26A6 phenocopied the similar lumen expansion defects with *NFAT5*knockout.

## Data Availability

All data generated or analyzed during this study are included in the manuscript and Appendix A.

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
