# Peer review of "Nuclear Factor of Activated T Cells-5 Regulates Notochord Lumenogenesis in Chordate Larval Development"

_ijms, 2022, doi:10.3390/ijms232214407_

Round 1
Reviewer 1 Report
The paper “Nuclear Factor of Activated T cells-5 regulates notochord lumenogenesis in chordate larval development” from Muchun He1 and coauthors investigates the role of the gene
Ci-NFAT5 in process of notochord formation. NFAT family is a gene family of transcription factors that have been associated to the cellular response to hypertonic stress. In this paper the involvement of this gene in lumen formation in notochord cells has been analyzed during development using tools to interfere with gene function like DN forms and CRISPR/Cas9 showing a phenotype compatible with a role for this gene In the process of tubu formation. Moreover a clear respons of Ci NFat5 to hypertonic stress has been demonstrated in an heterologous in vitro system. Finally ATAC seq has been used to found downstream genes regulated by Ci-NFAT5.
The manuscript is well written and the conclusions are well supported by the experiments.
One concern regards the lacking of in situ hybridization experiments: although the study done with promoters give evidence of expression in the notochord of the studied genes, in situ data could be very informative regarding the timing and the precise expression of the endogenouse gene. So I believe that the endogenouse expression pattern of NFAT5 during embryogenesis in Ciona is necessary.
Moreover a more detailed description of the experiments could is necessary as in the case of the DN experiments where the author wrote: To determine the roles of Ciona-NFAT5 in notochord lumen formation, we generated C-127 terminal deletion construct of Ciona NFAT5 fusion proteins (Ciona-NFAT5 DN), which 128 retains DNA binding domain and dimerization domain, serving as a dominant negative 129 mutant [31, 32] (Figure 2A), and forced it to express in notochord.
It is not clear which is the promoter used to do this experiment.
Minor points
Fig 3B: scale bar is lacking
Fig 3C is misleading with E
Author Response
Review 1
The paper “Nuclear Factor of Activated T cells-5 regulates notochord lumenogenesis in chordate larval development” from Muchun He1 and coauthors investigates the role of the gene Ci-NFAT5 in process of notochord formation. NFAT family is a gene family of transcription factors that have been associated to the cellular response to hypertonic stress. In this paper the involvement of this gene in lumen formation in notochord cells has been analyzed during development using tools to interfere with gene function like DN forms and CRISPR/Cas9 showing a phenotype compatible with a role for this gene In the process of tubu formation. Moreover a clear respons of Ci NFat5 to hypertonic stress has been demonstrated in an heterologous in vitro system. Finally ATAC seq has been used to found downstream genes regulated by Ci-NFAT5.
The manuscript is well written and the conclusions are well supported by the experiments.
Major concerns:
One concern regards the lacking of in situ hybridization experiments: although the study done with promoters give evidence of expression in the notochord of the studied genes, in situ data could be very regarding the timing and the precise expression of the endogenouse gene. So I believe that the endogenouse expression pattern of NFAT5 during embryogenesis in Ciona is necessary.
Response: Thanks for the suggestion. We agree that in situ hybridization results will be more accurate in determining the timing and the precise expression of Nfat5. The in situ experiments had been done in a previously published paper “The identification of transcription factors expressed in the notochord of Ciona intestinalis adds new potential players to the brachyury gene regulatory network”[1]. The results showed that Nfat5 was mainly expressed in the notochord from Neurula stage.
Major concerns:
Moreover a more detailed description of the experiments could is necessary as in the case of the DN experiments where the author wrote: To determine the roles of Ciona-NFAT5 in notochord lumen formation, we generated C-127 terminal deletion construct of Ciona NFAT5 fusion proteins (Ciona-NFAT5 DN), which 128 retains DNA binding domain and dimerization domain, serving as a dominant negative 129 mutant [31, 32] (Figure 2A), and forced it to express in notochord.
It is not clear which is the promoter used to do this experiment.
Response: We are sorry for not explaining this clearly. To determine the roles of Ci-NFAT5 in notochord lumen formation, we generated C-terminal deletion construct of Ciona NFAT5 fusion proteins (Ciona-NFAT5 DN), which retains DNA binding and dimerization domains, and lack the transactivation domain, serving as a dominant negative (DN) mutant (Figure 2A). This mutant construct had been reported to diminish hyperosmotic induction of the downstream gene in a dominant negative manner [2, 3]. Ciona-NFAT5 DN construct was electroporated into the fertilized eggs with a notochord-specific brachyury promoter to force it express in the notochord cells. We have added above text in the revision.
Minor points
Fig 3B: scale bar is lacking
Response: Thanks for the reminding. After checking, we found that scale bar is missing in Fig 2B. We have added the scale bar in the revised manuscript.
Fig 3C is misleading with E
Response: we don’t understand this point and didn’t find “E” in Fig 3C.
- Jose-Edwards, D.S., et al., The identification of transcription factors expressed in the notochord of Ciona intestinalis adds new potential players to the brachyury gene regulatory network. Dev Dyn, 2011. 240(7): p. 1793-805.
- Adachi, A., et al., NFAT5 regulates the canonical Wnt pathway and is required for cardiomyogenic differentiation. Biochem Biophys Res Commun, 2012. 426(3): p. 317-23.
- Ko, B.C., et al., Purification, identification, and characterization of an osmotic response element binding protein. Biochem Biophys Res Commun, 2000. 270(1): p. 52-61.

Reviewer 2 Report
This manuscript shows in a very clear and concise manner the formation of the notochord in the Urochordata group. The changes at the gene, molecular and cellular level, described in the paper, very illustratively show the changes that lead to the formation of the notochord.
The literature that has been used in the manuscript is adequate.
The review of the literature showed that all relevant literature directly related to the subject of the research was used.
The group on which the research was carried out is a group with a very complex and specific way of life, so both the design and the experiment were carried out in the best possible way.
The combined results of gene analysis (their sequences) and their phylogenetic analysis is very important for a better understanding of the functioning of the notochord formation process in all groups of chordates.
Results clealry showed that changes in osmotic pressure affect the localization of Ciona-NFAT5 protein in HeLa.
The interesting result of phylogenetic and expression pattern analysis of NFAT5 Ciona NFAT family genes showed that there are two clades: NFAT1-4 were clustered together while NFAT5 proteins were clustered in another clade.
NFAT1-4 proteins have calcineurin binding domain and were identified only in vertebrate animals, while NFAT5 proteins have no calcineurin binding domain and were identified in both invertebrate and vertebrate.
This result of analysis is important part of this work and can lead to better understanding of molecular level of early development stages of vertebrates.
After all, work is original; the results clearly indicate the role and importance of certain genes, metabolic processes and environmental factors on the formation of the notochord, which are only hinted at in similar works.
Author Response
Thanks very much for the comments from this reviewer.